

**Numerical modelling of flow and transport in Bari industrial area by means of**
**rough walled parallel plate and random walk models**
Claudia Cherubini[1,2], Nicola Pastore[3], Dimitra Rapti[4], Concetta I. Giasi[3]
[1]Department of Physics & Earth Sciences, University of Ferrara, Via Saragat 1- 44122, Ferrara (Italy)
[2]School of Civil Engineering, University of Queensland, St Lucia, Brisbane 4072, Australia;
[3]DICATECh, Department of Civil, Environmental, Building Engineering, and Chemistry, Politecnico di Bari,
Bari, Italy.
[4]New Energies And environment Company (NEA ) Via Saragat 1- 44122 Ferrara (Italy)
**Abstract**
Modelling fluid flow and solute transport dynamics in fractured karst aquifers is one of the
most challenging tasks in hydrogeology.
The present study investigates on the hotspots of groundwater contamination in the industrial
area of Modugno (Bari –Southern Italy) where the limestone aquifer has a fractured and karstic
nature.
A rough walled parallel plate model coupled with a geostatistical analysis to infer the values of
the equivalent aperture has been implemented and calibrated on the basis of piezometric data.
Using the random walk theory, the steady state distribution of hypothetical contamination with
the source at the hot spot has been carried out reproducing a pollution scenario which is
compatible with the observed one. From an analysis of the flow and transport pattern it is
possible to infer that the anticline affecting the Calcare di Bari formation in directions ENE-
WSW influences the direction of flow as well as the propagation of the contaminant.
The results also show that the presence of nonlinear flow influences advection, in that it leads
to a delay in solute transport respect to the linear flow assumption. This is due to the not constant
distribution of solute according to different pathways for fractured media which is related to
the flow rate.
**Introduction**
The characterization and the description of phenomena that involve fractured aquifers,
especially if considered in relationship with water resource exploitation, is an important issue
because fractured aquifers serve as the primary source of drinking water for many areas of the
world. In fractured rock aquifers, groundwater is stored in the fractures, joints, bedding planes
and cavities of the rock mass. Water availability is largely dependent on the nature of the
fractures and their interconnection. Fractures enable fast pathways for fluid flow that can



transport contaminants. The ability of a fracture to transmit water as well as contaminants
depends primarily on the size of the opening, or the fracture aperture.
The parallel plate model is widely used to simulate flow in a fracture due to its simplicity of
idealizing a fracture. Many workers (Baker, 1955; Huitt, 1956; Snow, 1968, 1970; Gale, 1977)
have used flow between smooth parallel plates as a model for flow in fractures. The solution to
the Navier-Stokes equation for flow between parallel plates, known as plane Poiseuille flow,
has been known to fluid mechanicians since the nineteenth century.
Witherspoon et al. (1980) and Elliott et al. (1985) suggested that a factor should be introduced
into the parallel plate theory to take account of the effects of joint surface properties.
Zhao and Brown (1992) carried out hydro-thermo-mechanical tests on joints in the
Carnmenellis granite from Cornwall, southwest England using a geothermal rock test facility.
Experimental effective normal stress-joint closure and effective normal stress-joint
permeability data were fitted by a range of deformation and hydraulic models. They applied the
joint condition factor (JCF) to account for deviations from the ideal condition assumed in the
smooth parallel plate theory reflecting the effects of joint roughness, joint matching, joint
stiffness, deposits of detritus, loading history, sample disturbance, sample size and the
temperature environment.
Zimmermann and Bodvarsson (1996) discussed the problem of fluid flow through a rock
fracture within the context of fluid mechanics. The derivation of the 'cubic law' was given as
the solution to the Navier-Stokes equations for flow between smooth, parallel plates. They
analysed the various geometric and kinematic conditions that are necessary in order for the
Navier-Stokes equations to be replaced by the more tractable lubrication or HeleShaw equations
and reviewed various analytical and numerical results pertaining to the problem of relating the
effective hydraulic aperture to the statistics of the aperture distribution.
They found that the effective hydraulic aperture is less than the mean aperture, by a factor that
depends on the ratio of the mean value of the aperture to its standard deviation. Finally, they
compared the predicted hydraulic apertures to measured values for eight data sets from the
literature for which aperture and conductivity data were available on the same fracture. They
concluded that reasonably accurate predictions of hydraulic conductivity can be made based
solely on the first two moments of the aperture distribution function, and the proportion of
contact area.
Some researchers proposed a variable aperture model stating that it is better adapted to describe
flow and transport channeling effects than a parallel plate model (Neretnieks et al., 1982;
Bourke, 1987; Pyrak-Nolte, 1988; Tsang and Tsang, 1989; Tsang et al., 2001) where fracture



apertures can be described by normal, (Lee et al., 2003), lognormal (e.g., Keller, 1998; Keller
et al., 1999), or gamma distributions (Tsang and Tsang, 1987), or a self-affine scale invariance
(Plouraboue et al., 1995).
Neuzil and Tracy (1981) presented a model for flow in fractures where the flow is envisioned
as occurring in a set of parallel plate openings with different apertures whose distribution was
lognormal and used a modified Poiseuille equation.
They showed that the flow conformed to the cubic law and also that the maximum flow occurs
through the largest apertures, thereby emphasizing that flow occurs through preferred paths.
Thus in their analysis, the flow depended on the tail of the frequency distribution.
Tsang and Tsang, (1987) proposed a theoretical approach to interpret flow in a tight fractured
medium in terms of flow through a system of statistically equivalent one-dimensional channels
of variable aperture. The channels were statistically equivalent in the sense that the apertures
along each flow channel are generated from the same aperture density distribution and spatial
correlation length.
Oron and Berkowitz (1998) have examined the validity of applying the 'local cubic law' (LCL)
to flow in a fracture which is bounded by impermeable rock surfaces. A two- dimensional order-
of-magnitude analysis of the Navier-Stokes equations yields three conditions for the
applicability of LCL flow, as a leading-order approximation in a local fracture segment with
parallel or nonparallel walls. These conditions demonstrate that the 'cubic law' is valid provided
that aperture is measured not on a point-by-point basis but rather as an average over a certain
length.
Experimental work by Plouraboué et al. (2000) in self-affine rough fractures with various
translations of the opposing fracture surfaces indicated that heterogeneity in the flow field
caused deviations from the parallel plate model for fracture flow.
Some researchers often find it convenient to represent aperture fields in terms of equivalent
aperture in the parallel plate model (Zheng et al., 2008).
Zheng et al., 2008 carried out a systematic series of hydraulic and tracer tests on three
laboratory-scale fracture replicas, and calculated the cubic law, mass balance, and frictional
loss apertures. They fitted an analytical solution to the one-dimensional advection-dispersion
equation to each experimental breakthrough curve three times, each time applying v based on
one of the three ''equivalent apertures''.
The excellent agreement between the experimental breakthrough curves and the simulated
curves based on the single-parameter curve fit applying the mass balance aperture clearly




demonstrates that the mass balance aperture is the only equivalent aperture appropriate for
describing solute transport in single variable-aperture fractures.
Brush and Thomson (2003) developed three-dimensional flow models to simulate fluid flow
through various random synthetic rough-walled fractures created by combining random fields
of aperture and the mean wall topography or midsurface, which quantifies undulation about the
fracture plane.
The total flow rate from three-dimensional Stokes simulations were within 10% of LCL
simulations with geometric corrections for all synthetic fractures. Differences between the NS
and Stokes simulations clearly demonstrated that inertial forces can significantly influence the
internal flow field within a fracture and the total flow rate across a fracture.
Klimczak e a. (2010) carried out flow simulations through fracture networks using the discrete
fracture network model (DFN) where flow was modeled through fracture networks with the
same spatial distribution of fractures for correlated and uncorrelated fracture length-to-aperture
relationships. Results indicate that flow rates are significantly higher for correlated DFNs.
Furthermore, the length-to-aperture relations lead to power-law distributions of network
hydraulic conductivity which greatly influence equivalent permeability tensor values. These
results confirm the importance of the correlated square root relationship of displacement to
length scaling for total flow through natural opening-mode fractures and, hence, emphasize the
role of these correlations for flow modeling.
Wang et al. (2015) developed and tested a modified LCL (MLCL) taking into account local
tortuosity and roughness, and works across a low range of local Reynolds Numbers. The MLCL
was based on (1) modifying the aperture field by orienting it with the flow direction and (2)
correcting for local roughness changes associated with local flow expansion/contraction. In
order to test the MLCL, they compared it with direct numerical simulations with the Navier-
Stokes equations using real and synthetic three-dimensional rough-walled fractures, previous
corrected forms of the LCL, and experimental flow tests. The MCL proved to be more accurate
than previous modifications of the LCL.
The CTRW approach provides a versatile framework for modelling (non-Fickian) solute
transport in fractured media.
Berkowitz et al (2001) examined a set of analytical solutions based on the continuous time
random walk (CTRW) approach to analyze breakthrough data from tracer tests to account for
non-Fickian (or scale-dependent) dispersion behavior that cannot be properly quantified by
using the advection-dispersion equation.



Cortis et al. (2008) developed a macroscopic model based on the Continuous Time Random
Walk (CTRW) framework, to characterize the interaction between the fractured and porous
rock domains by using a probability distribution function of residence times. They presented a
parametric study of how CTRW parameters evolve, describing transport as a function of the
hydraulic conductivity ratio between fractured and porous domains.
Srinivasan et al. (2010) presented a particle-based algorithm that treats a particle trajectory as
a subordinated stochastic process that is described by a set of Langevin equations, which
represent a continuous time random walk (CTRW). They used convolution based particle
tracking (CBPT) to increase the computational efficiency and accuracy of these particle-based
simulations. The combined CTRW–CBPT approach allows to convert any particle tracking
legacy code into a simulator capable of handling non-Fickian transport.
Dentz et al (2016) developed a general CTRW approach for transport under radial flow
conditions starting from the random walk equations for the quantification of non-local solute
transport induced by heterogeneous flow distributions and by mobile-immobile mass transfer
processes. They observed power-law tails of the solute breakthrough for broad distributions of
particle transit times and particle trapping times. The combined model displayed an
intermediate regime, in which the solute breakthrough is dominated by the particle transit times
in the mobile zones, and a late time regime that is governed by the distribution of particle
trapping times in immobile zones.
The present study is aimed at analysing the scenario of groundwater contamination of the
industrial area of Modugno (Bari –Southern Italy) where the limestone aquifer has a fractured
and karstic nature.
Previous studies carried out in the same aquifer have applied different conceptual models to
model fluid flow and contaminant transport.
Cherubini (2008) applied the discrete feature approach (Diersch, 2002) where the 3D geometry
of the subsurface domain describing the matrix structure was combined by interconnected 2D
and 1D discrete feature elements in two dimensions in order to simulate respectively fractures
and karstic cavities in the Bari limestone aquifer. The fracture distribution was inferred from a
nonparametric geostatistical analysis (Indicator Kriging) of fracture frequency data which had
been derived by RQD (Priest and Hudson, 1976) data of the contaminated area of the ex
Gasometer.
Cherubini et al. (2008) compared the flow modelling results of the previous work with those
from a new hydrogeological reconstruction of the heterogeneities in the same aquifer by means



of multiple realizations conditioned to borehole data (RQD population), in order to obtain a
three-dimensional distribution of fracture frequency, cavities and terra rossa lenses.
Cherubini and Pastore (2010) applied the nested sequential indicator simulation algorithm to
represent the geological architecture of the Bari limestone aquifer which provided realiable
prediction of fluid flow. According to phenols transport, the presence of preferential pathways
was detected.
Cherubini et al. (2013) realised a 3D flow model of Bari limestone aquifer supported by a
detailed local scale geologic model realised by means of Sequential indicator simulation (SIS)
of lithofacies unit sequences. In this study, a lumped parameter approach was used and
calibrated on the groundwater discharge and global hydraulic gradient where fluid flow in
fractures was represented by the cubic law, and Darcy–Weisbach equation was used to estimate
resistance term in karst network.
Masciopinto et al. (2010) adopted a conceptual model consisting of a 3D parallel set of
horizontal planar fractures in between rock layers, each fracture having a variable aperture
generated by a stationary random field conditioned to the data derived from pumping-tracer
tests. The particle tracking solution was combined with the PHREEQC-2 results to study two-
dimensional laminar/non-laminar flow and reactive transport with biodegradation in each
fracture of the conceptual model.
Masciopinto and Palmiotta (2013) derived new equations of fracture aperture as functions of a
tortuosity factor to simulate fluid flow and pollutant transport in fractured aquifers.
MODFLOW/MT3DMS water velocity predictions were compared with those obtained using a
specific software application which solves flow and transport problems in a 3D set of parallel
fissures. The results of a pumping/tracer test carried out in a fractured limestone aquifer in Bari
(Southern Italy) have been used to calibrate advective/dispersive tracer fluxes given by the
applied models. Successful simulations of flow and transport in the fractured limestone aquifer
were achieved by accommodating the new tortuosity factor in models whose importance lies in
the possibility of switching from a discrete to a continuum model by taking into account the
effective tracer velocity during flow and transport simulations in fractures.
Masciopinto and Visino (2017) carried out filtration tests on a set of 16 parallel limestone slabs
having a thickness of about 1 cm where rough surfaces and variable fracture apertures had been
artificially created. The experimental filtration results suggest that model simulations of
perturbed virus transport in fractured soils need to also consider also pulse-like sources and
sinks of viruses. This behavior cannot be simulated using conventional model equations without
including a new kinetic model approach.


The present work focuses on the investigation of the hotspots of aquifer contamination in order
to infer the location of the sources.
A rough walled parallel plate model has been implemented and calibrated on the basis of
piezometric data and has coupled a geostatistical analysis to infer the values of the equivalent
aperture.

## Geological and hydrogeological framework

It is well known that hydraulic properties and consequently fluid circulation and contaminant
propagation in carbonate rocks are strongly influenced by the degree of rock fracturing and, in
general, the presence of mechanical discontinuities, like faults, joints, or other tectonic elements
such as syncline or anticline axes (Caine et al., 1996; Caine and Foster, 1999; Antonellini et
al., 2014; Billi et al., 2003). Also, the deformation mechanisms are mainly controlled by the
physico-chemical properties of rocks, which are, in turn, the result of different composition,
depositional setting and diagenetic evolution (Zhang and Spiers, 2005; Rustichelli et al., 2012).
From the geological point of view, the investigated area is located in the Murge Plateau
corresponding to a broad antiformal structure oriented WNW- ESE and represents the bulging
foreland of the Pliocene-Pleistocene Southern Apennines orogenic belt (Pieri et al., 1997;
Doglioni, 1994; Foster and Evans, 1991; Korneva et al., 2014; Parise and Pascali, 2003).
The stratigraphy of the Murge area consists of a Variscan crystalline basement topped by 6-7
km-thick Mesozoic sedimentary cover (represented by the Calcare di Bari formation) followed
by relative thin and discontinuous Cenozoic and Quaternary deposits (Calcareniti di Gravina
formation). Figure 1 shows the simplified geological map of the area of Bari

*Calcare di Bari formation (Cretaceous)*
The Calcare di Bari succession consists of biopeloidal and peloidal wackestones/packstones
alternating with stromatolithic bindstones with frequent intercalations of dolomitic limestones
and grey dolostones. The formation
shows a thickness of about 470 m. Most of the Calcare di Bari formation shows facies features
related to peritidal environments; only the upper part suggests a relatively more distal and
deeper environment belonging to an external platform setting (CARG project, 2010; Fig.1).
This succession appears stratified and fissured and, where it is not interested by tectonical
discontinuities, it shows a subhorizontal or slightly inclined lying position. This formation is
subjected to the complex and relevant karstic phenomena that, locally lead to the formation of
cavities of different shapes and sizes, partially or completely filled by "terra rossa" deposits.





On the basis of borehole and in situ surveys, carried out by private companies it was observed
that:
-    The fracturing degree of the Calcare di Bari formation is quite variable and it is
expressed by Rock Quality Designation (RQD) values that vary between 16 and 25%
(maximum borehole depths: 30 m).  Based on the classification system of Deere and Deere
(1988) the rock mass is of 'very poor rock quality' (RQD <25%).
-    Medium values of Rock Mass Rating (RMR) about 36, indicate, after Bieniawski
classification (1989), very poor rock mass (class IV).
-    In addition, profiles of the electrical resistivity (depth < 30 m) have permitted to observe
the presence of very variable electric resistivity values with variations between 100 (low
fracture carbonates rocks) and 1700 Ohm*m for very fractured formations; with locally values
of the order of 3-4000 Ohm*m, in case of underground cavities.
-    Similarly, the velocity of the seismic waves P and S has average values of the order of
1300 and 800 m/s, respectively in highly fractured limestones and 2300 (P) and 1400 (S) m/s
for compact formations.

*Calcareniti di Gravina formation (lower Pleistocene)*
This unit uncomfortably lies on the Calcare di Bari Fm.  Its thickness varies from few meters
to 20 m and its depositional environments are related to offshore setting. The lower boundary
is transgressive and is locally marked by reddish residual deposits and/or by brackish silty
deposits passing upward to shallow water calcarenites rich in bioclasts.
As regards the structural features of these deposits it is possible to observe the anticline
affecting only the Cretaceous succession of the Calcare di Bari formation in directions ENE-
WSW. This tectonic structure influences the direction of flow as shown in Fig. 1. Also, the
presence of this fault line with direction NE-SW, controls the development of the actual
hydrographic network.
The limestone bedrock hosts a wide and thick aquifer. The intense fracturing of the rock and
the karst phenomenon result in a high permeability of the limestone where the groundwater
flows primarily through the fractures and open joints.
Moreover, the irregular spatial distribution of the fractures and karstic channels renders the Bari
aquifer very anisotropic. The hydraulic conductivity of this aquifer is generally estimated in
$10^{-3}$ to $10^{-4}$ m/s.
The groundwater flows toward the sea, under a low pressure, in different subparallel fractured
layers separated by compact (i.e., not fractured) rock blocks.



In proximity to the coast, the carbonate (Mesozoic) stratum contains fresh water flowing in
phreatic conditions and floating on underlying saltwater of continental intrusion. The location
of the transition zone between fresh and salt water has thickness and position variable and
changes over time depending on the distribution of the hydrostatic pressures of the system.

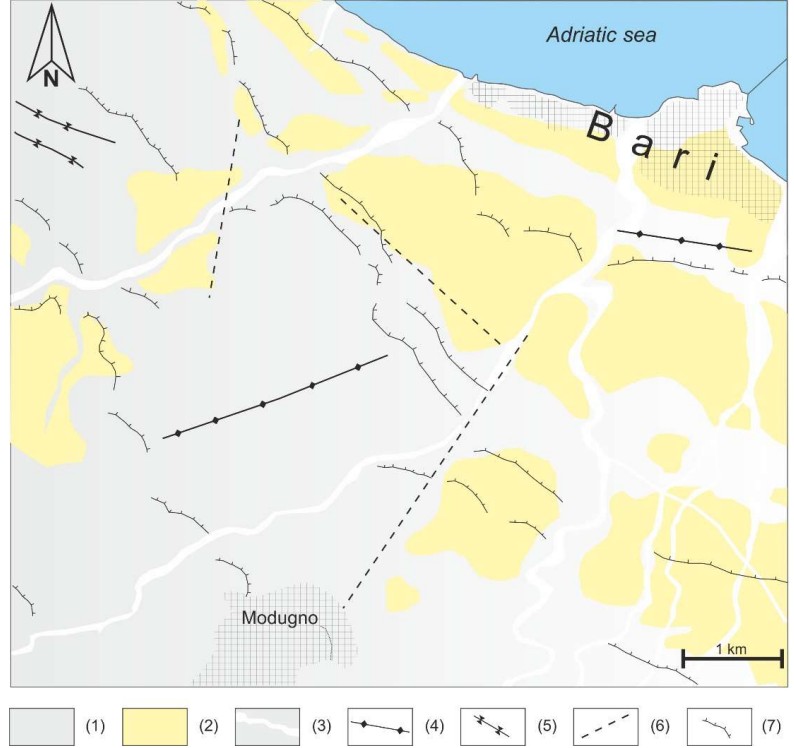



**Figure 1. Simplified geological map of the area of Bari: (1) Calcare di Bari formation (Cretaceous); (2) Calcareniti di**
**Gravina formation (Lower Pleistocene); (3) hydrographic network; (4) anticlinal axis; (5) syncline axis; (6) fault**
**(uncertain); (7): escarpment.**

**Hydrologic and hydrogeologic water budget**
By means of the hydrologic and hydrogeologic water budget of the subtended basin the
effective infiltration has been estimated.
Climatic data registered in the thermopluviometric stations present in the area have been
elaborated and the average rainfall module and the monthly evapotranspiration have been
calculated for the three decades 1974-2005.
12 climatic stations have been considered (Bari – hydrographic station, Bari – observatory
station, Bitonto, Grumo appula, Adelfia, Casamassima, Mercadante, Ruvo di Puglia, Corato,
Altamura, Santeramo, Gioia del Colle) and for each station and the monthly rainfall and



evapotranspiration map has been realised by means of the *Inverse distance weighting* algorithm.
The latter has been estimated by means of Thornthwaite method applying a crop coefficient of

291    0.40.

The hydrologic and hydrogeologic basin have been defined on the basis of literature data and
the regional thematic cartography.
The lithotypes in the study area are principally limestones and calcarenites with secondary
permeability, characterised by a high transmissivity. The zones in proximity of tectonic
structures create preferential flow paths but at the same time generate a dismemberment of the
aquifer that could not be able to feed the flow downstream. Because of that it proves to be
difficult to carry out a zonation of recharge areas and therefore a constant run off coefficient of
0.10 has been considered for the whole basin. In Figure 2 the map of the a) annual precipitation
and b) real evapotranspiration evaluated for the hydrological basin of the study area is shown.

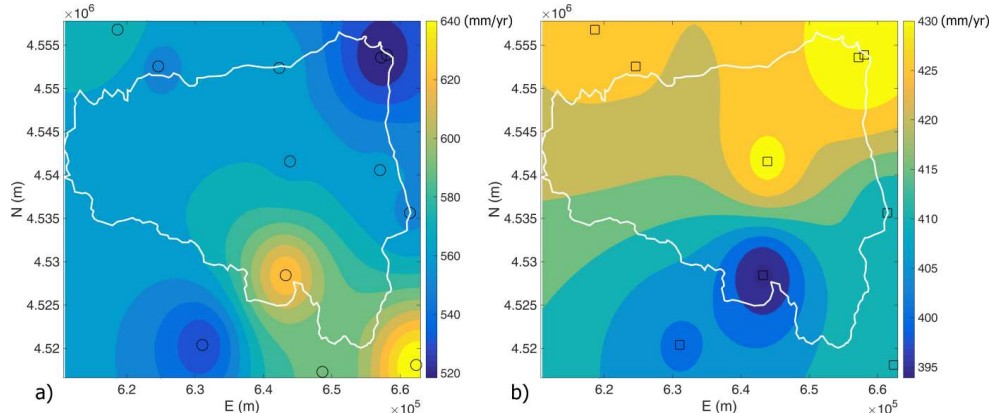


**Figure 2. Map of a) annual precipitation b) real evapotranspiration evaluated for the hydrological basin of the study**
**area.**

**Well performance tests: step-drawdown tests**
98 long term step drawdown hydraulic tests have been analysed in the study area.
A step-drawdown test is a single-well test in which the well is pumped at a low constant
discharge rate until the drawdown within the well stabilizes.
Step drawdown tests can be used to evaluate the characteristics of the well and its immediate
environment. Unlike the aquifer test, it is not designed to produce reliable information
concerning the aquifer, even though it is possible to estimate the transmissivity of the immediate
surroundings of the catchment. This test determines the critical flow rate of the well, as well as
the various head-losses and drawdowns as functions of pumping rates and times. Finally, it is





designed to estimate the well efficiency, to set an exploitation pumping rate and to specify the
depth of installation of the pump.
The total drawdown at a pumping well is given by:
$$s = (A_1 + A_2) \cdot Q + B \cdot Q^2 \qquad (1)$$
Where $s$ (L) represents the registered drawdown, $Q$ ($L^3T^{-1}$) the pumped flow rate, $A_1$ ($TL^{-2}$) is
the linear aquifer loss coefficient, $A_2$ ($TL^{-2}$) e $B$ ($T^2L^{-5}$) = are respectively the linear and
nonlinear well-loss coefficients.
This equation can be explicited in terms of aquifer transmissivity T ($L^2T^{-1}$), the transmissivity
of damage zone $T_{SKIN}$ ($L^2T^{-1}$) and of the nonlinear term $\beta$ ($T^2L^{-4}$) (Cherubini & Pastore, 2011):
$$s = \left[ \frac{1}{T2\pi} \ln\left( \frac{R}{r_w} \right) + \frac{1}{2\pi} \left( \frac{1}{T_{SKIN}} - \frac{1}{T} \right) \ln\left( \frac{r_{SKIN}}{r_W} \right) \right] Q + \left[ \frac{\beta}{4\pi^2} \left( \frac{1}{r_w} - \frac{1}{R} \right) \right] Q^2 \qquad (2)$$
Where $r_w$ (L) represents the well radius, $r_{SKIN}$ (L) the radius of the damage zone, R (L) the radius
of influence of the well.
The total drawdown is formed of three components: the hydraulic component of the aquifer
assuming valid Thiem function, a skin function presented by Cooley and Cunningham (1979)
assuming that the transmissivity and the radius of the damage zone are respectively equal to:
$T_{SKIN} = T / 2$  e  $r_{SKIN} = 2r_w$; and a contribution related to nonlinear losses introduced by Wu

330 (2001).

The radius of influence of the well is obtained by means of Sichart equation:
$$R = 3000 \cdot s \cdot \sqrt{K} \qquad (3)$$
In figure 3 is reported the statistical distribution of the estimated transmissivity values along
the study area.





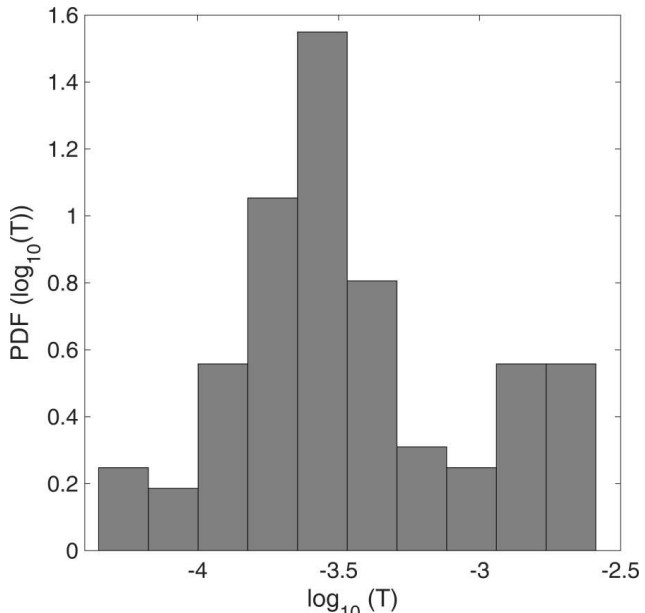


**Figure 3. Statistical distribution of log$_{10}$ (T).**

**Linear model of regionalization of Transmissivity**
The geostatistical analysis has been carried out on the log$_{10}$ transmissivity values using the open
source code S-GemS (Remy, 2004).
The experimental variogram, which provides a description of how the data are related
(correlated) with distance, has been calculated (Figure 4). Because the kriging algorithm
requires a positive definite model of spatial variability, the experimental variogram cannot be
used directly. Instead, a model must be fitted to the data to approximately describe the spatial
continuity of the data. An exponential model has been used to fit the experimental variogram
described by the function:
$$\gamma\left(h\right)=C\left[1-\exp\left(-\frac{h}{a}\right)\right] \tag{4}$$
Where $C$ represents the variance (sill), $h$ [L] the lag and $a$ [L] the correlation length (range). In
our case $C$ assumes a value of 1.2 and $a$ of 10000 m.





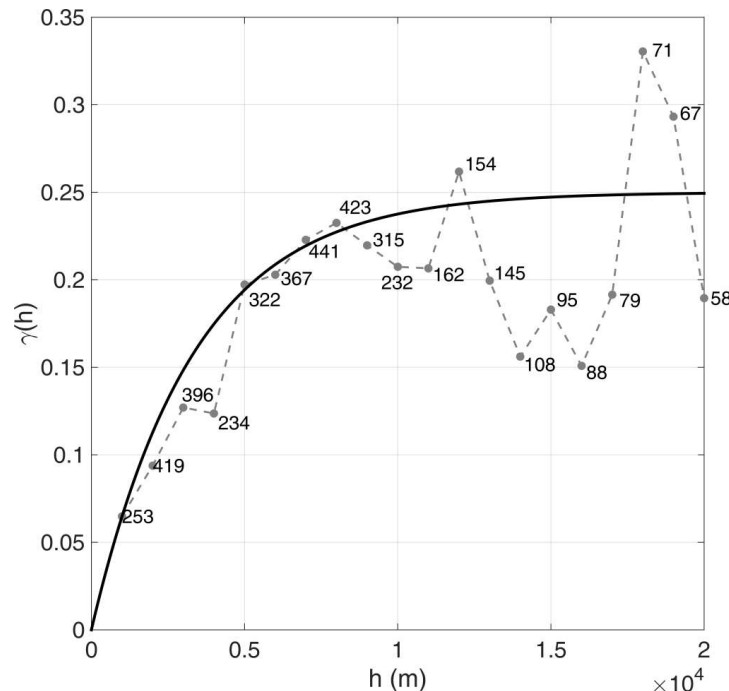


**Figure 4. Omnidirectional experimental variogram fitted with an exponential model, sill = 1.2, range = 10000 m.**


Figure 5 shows the ordinary Kriging interpolation of $\log_{10}(T)$.





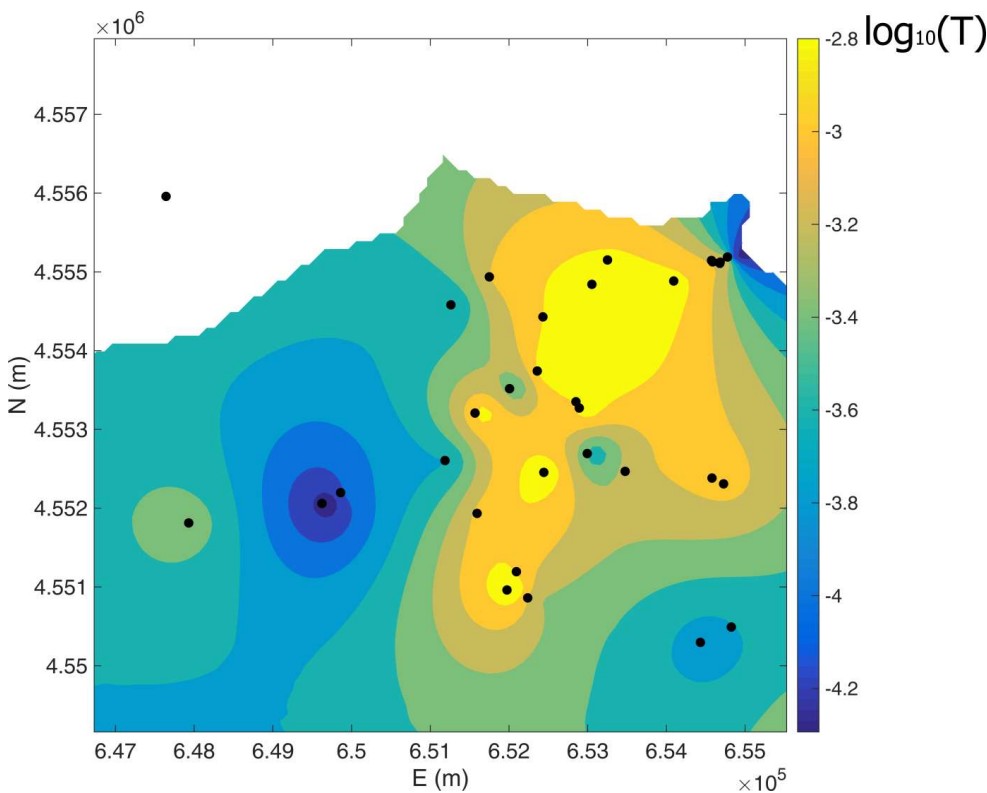


**Figure 5. Ordinary Kriging interpolation of log$_{10}$(T).**

**Analysis of piezometric data**
Figure 6 shows the spatial distribution of hydraulic heads on the basis of 2012 sampling
campaign. A global trend in the direction of groundwater flow from SW to NE is evident. A
relevant aspect is the presence of high hydraulic head values in proximity of ASI and Bosch
wells. A possible explanation for this could be the presence of a zone of poor connection of
groundwater flow patterns in correspondence of that zone. The aquifer transmissivity in that
zone is of the order of $10^{-5}$ m/s. The trend observed in the hydraulic gradient confirms the
increase of the aquifer transmissivity from upstream to downstream, in fact the tests carried out
in proximity of the coast have returned a transmissivity value of $10^{-2}$ m/s.





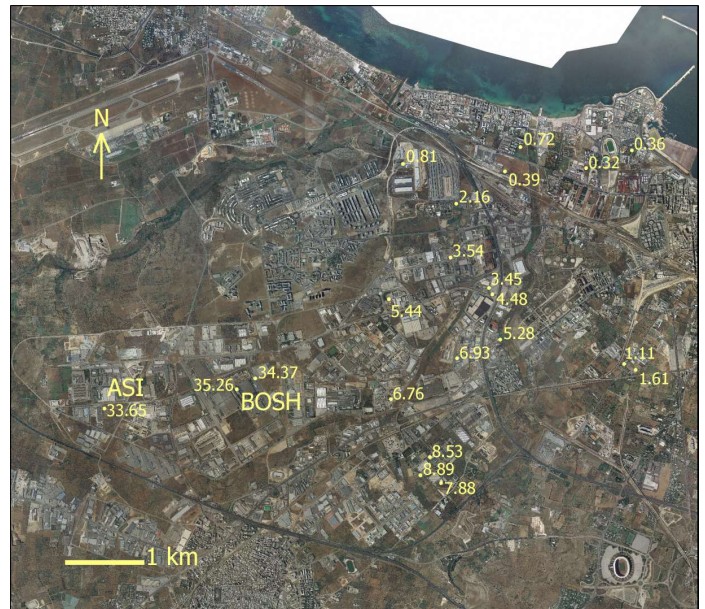

**367**

**368** **Figure 6. Measured piezometric heads (m, slm) from February 2012 monitoring campaign**

**369**

**370** **Analysis of the scenario of contamination for the study area**

**371** The various monitoring campaigns carried out have showed a contamination of Chlorinated

**372** Aliphatic Hydrocarbons which, unlike petroleum products, are denser than water and can exist

**373** as Dense Non-Aqueous Phase Liquids (DNAPLs).

**374** The presence of two hot spot areas has been detected, located upstream of the groundwater

**375** flow, coherently with the state of contamination detected downstream.

**376** Figure 7 shows the location of the detected contamination (µg/l).

**377** The pollution indicator has been chosen on the basis of the toxicologic and cancirogenic

**378** parameters, the solubility, the sorption coefficient and the maximum detected contaminant

**379** concentration. On the basis of the results of this screening the Tetrachloroethylene (PCE) has

**380** the highest concentration as well as low values of Reference Dose Factors (RfD) and Slope

**381** Factors (SF).






**Figure 7. Location of the detected contamination (µg/l).**

**Parallel rough-walled fracture model**
The simplest model of flow through rock fractures is the parallel plate model (Huit, 1955; Snow,
1965) which conceptualises the fractured medium as made by a set of smooth parallel plates
having the same hydraulic aperture $b_{eq}$ (L) that are separated by a uniform distance. This is
actually the only geometrical fracture model for which an exact calculation of the hydraulic
conductivity is possible.
Natural fractures present rough walls and complex geometries. Nonlinear flow may occur
through rough-walled rock fractures as a consequence the inertial effect dominate the flow
dynamics giving rise a deviation from darcy's law. Fluid flow through a set of natural fracture
planes can be expressed using the Darcy-Weisbach equation:
$$\frac{dh}{dx} = -\frac{f}{D}\frac{v^2}{2g}$$ (5)
Where $D$ (L) represents the hydraulic diameter (2b for the parallel plate model), $f$ the Darcy –
Weisbach coefficient, $h$ (L) is the hydraulic head, $x$ (L) is the distance and $v$ (LT$^{-1}$) is the average
velocity in fracture calculated as:
$$v = \frac{q}{n_f b}$$ (6)
Where $q$ (L$^2$T$^{-1}$) is the volumetric flow rate per unit length of fractures.



The Darcy – Weisbach equation can be rewritten in terms of volumetric flow per unit length:
$$\overline{q} = -\left[ n_f b \frac{\sqrt{\frac{4b}{f} g}}{\sqrt{|\nabla h|}} \right] \nabla h \qquad (7)$$

The term in square bracket represents the equivalent hydraulic transmissivity $T_{eq}(f, \nabla h)$ of the
$n_f$ rough - walled fractures.
The Darcy-Weisbach coefficient or friction factor depends of the flow regime. In the case of
smooth-walled fracture and linear flow regime $f$ is equal to:
$$f = \frac{96}{\text{Re}} \qquad (8)$$

Where Re represents the Reynolds number:
$$\text{Re} = \frac{\rho v D}{\mu} \qquad (9)$$

Substituting equation (8) in equation (7) the cubic law (Witherspoon et al., 1980) where q is
proportional to the cubic power of the fracture aperture is obtained:
$$q = n_f \frac{\rho g}{\mu} \frac{b^3}{12} \qquad (10)$$

The cubic law is not always adequate to represent the flow process in natural fractures, a
deviation from linearity can be observed.
The friction factor depends from the flow regime described by the Reynolds number and can
be presented with the following relationship found by Nazridoust at al. (2006):
$$f = \frac{123}{\text{Re}} \left( 1 + 0.12 \, \text{Re}^{0.687} \right) \qquad (11)$$


**Inverse flow modeling**
Inverse modelling is a technique used to estimate unknown model parameters using as input
data punctual values of the state variables (hydraulic head, flow). Generally, in real problems
the number of parameters to estimate (*n*) is higher than the number of measured values (*m*). For
example, this is the case of mapping hydraulic transmissivity values varying continuously in
space.
For underdetermined inverse problems of this kind the objective function (*L*) can be written in
this way:
$$L(\mathbf{y}, \mathbf{s}) = L_{fitness}(\mathbf{y}, \mathbf{s}) + L_{penalty}(\mathbf{s}) \qquad (12)$$




Where **s** represents the vector of measured values of state variables (es. hydraulic
transmissivity), **y** represents the vector of parameter values.
The *fitness function* responds to maximum likelihood criteria between the observed and the
simulated values and can be written as:
$$L_{fitness}(\mathbf{y},\mathbf{s}) = (\mathbf{y} - \mathbf{h(s)})^T \mathbf{R}^{-1}(\mathbf{y} - \mathbf{h(s)})$$ (13)
Where **h** represents the model that, starting from the parameter vector, estimates the state
variable, **R** is the measurement error covariance matrix. Generally this function can be reduced
to the square root of the sum of the squared difference between the measured and simulated
(RMSE):
$$L_{fitness} = \frac{\|\mathbf{y} - \mathbf{h(s)}\|^2}{\Delta H^2}$$ (14)
Where ΔH represents a parameter of accuracy of observed data.
The *penalty function* is used to discriminate the solutions with values of the fitness function
comparable by means of geostatistical criteria (Kitanidis, 1995):
$$L_{penalty} = (\mathbf{s} - \mathbf{X}\beta)^T \mathbf{Q}^{-1}(\mathbf{s} - \mathbf{X}\beta)$$ (15)
Where **Q** represents the spatial covariance matrix, **X** is a unit vector and $\beta$ is the mean of the
values of the parameters. The penalty function can be rewritten eliminating $\beta$:
$$L_{penalty} = \mathbf{s}^T\mathbf{G}\mathbf{s} \qquad \mathbf{G} \equiv \mathbf{Q}^{-1} - \frac{\mathbf{Q}^{-1}\mathbf{X}\mathbf{X}^T\mathbf{Q}^{-1}}{\mathbf{X}^T\mathbf{Q}^{-1}\mathbf{X}}$$ (16)
The common assumption is that the spatial distribution of the parameters follows the
geostatistical distribution defined by the variogram. Under this hypothesis the covariance
matrix present in the penalty function can be defined as:
$$Q_{ij} = 2\gamma(|x_i - x_j|) \quad i,j = 1,....,n$$ (17)

**Solute transport modeling**
Solute transport in fracture neglecting the effect of matrix diffusion and the chemical reactions
can be described by the following advection dispersion equation:
$$\frac{\partial c}{\partial t} + \bar{v} \cdot \nabla c = \nabla \cdot (\mathbf{D}\nabla c)$$ (18)
Where $c$ (ML$^{-3}$) is the concentration of solute and **D** (L$^2$T$^{-1}$) is the symmetric dispersion tensor
having the following components:



$$Dxx = \left(\alpha_L v_x^2 + \alpha_T v_y^2\right)/|v|$$

$$Dyy = \left(\alpha_T v_x^2 + \alpha_L v_y^2\right)/|v| \tag{19}$$

$$Dxy = (\alpha_L - \alpha_T)v_x v_y /|v|$$

Where $\alpha_L$ (L) and $\alpha_T$ (L) are the longitudinal and transverse dispersion coefficients respectively.
In order to solve the advective transport equation a numerical Lagrangian particle based random
walk method is implemented. The solute plume is discretized into a finite number of particles.
For pure advective transport the particle moves along the flow lines. In order to represent
dispersion phenomena, the random walk method adding a random displacement to each
particle, independently of the other particles, in addition to advective displacement.
For a given time step $\Delta t$, considering the tensorial nature of the dispersion and the spatially
variable velocity field each, particle moves according to:
$$x_p\left(t+\Delta t\right) = x_p\left(t\right) + v'_x \Delta t + Z_1\sqrt{2D_L\Delta t}\frac{v_x}{|v|} - Z_2\sqrt{2D_T\Delta t}\frac{v_y}{|v|}$$

$$y_p\left(t+\Delta t\right) = y_p\left(t\right) + v'_y \Delta t + Z_1\sqrt{2D_L\Delta t}\frac{v_y}{|v|} + Z_2\sqrt{2D_T\Delta t}\frac{v_x}{|v|} \tag{20}$$

With:
$$v'_x = v_x + \frac{\partial D_{xx}}{\partial x} + \frac{\partial D_{xy}}{\partial y}$$

$$v'_y = v_y + \frac{\partial D_{xy}}{\partial x} + \frac{\partial D_{yy}}{\partial y} \tag{21}$$

$$D_L = \alpha_L |v|$$

$$D_T = \alpha_T |v|$$

For steady – state flow and for a source constant intensity, the assumption that the particles $N$
released in time interval $(t_1, t_1 + \Delta t)$ follow exactly the same random trajectories of the particles
$N$ released during the previous interval $(t_1, t_1 - \Delta t)$ is possible. Under this assumption only $N$
particles are needed to simulate the location of the particles at previous time step.

**Flow modeling**
The numerical code MODFLOW coupled with the inverse model approach presented in the
previous section has been used to model groundwater flow.
The numerical simulations have been carried out on a two-dimensional domain of 968.7 Km$^2$.
The domain has been discretised by means of a structured grid of 100 m size with.
In correspondence of the coast line a first type boundary condition has been imposed (h = 0 m),
along the detected watershed a second type boundary condition (q = 0 m$^2$/s), the recharge from



upstream is simulated by means of a first type boundary condition where the hydraulic heads
are equal to the detected regional values h = 32 – 41 m (Piano di Tutela delle Acque Regione
Puglia, Tav. 6.2 http://old.regione.puglia.it/index.php?page=documenti&id=29&opz=getdoc).
A second type boundary condition on the whole simulation domain has been imposed that
concerns the mean effective infiltration calculated from the hydrologic budget q = 0.037 md$^{-1}$.
The algorithm of inverse modelling has been applied to carry out the estimation of the spatial
distribution of the equivalent transmissivity (Figure 8) on the basis of the observed hydraulic
head (vector **y**), the regionalization model (matrix **Q**) described by the variogram of the
logarithmic of the hydraulic transmissivity determined in the previous section.
The inverse model algorithm follows those steps. 1) Starting from a conditional simulation of
the log of $T_{eq}$ determined by means of the hydraulic tests conducted in the area. 2) A set of pilot
points are chosen in the area using a regular spaced criteria and the value of $T_{eq}$ has been
determined for each pilot points (vector **s**). 3) By means of the Ordinary Kriging interpolation
of the pilot points the map of $T_{eq}$ is obtained and represents the input datum of the flow
numerical model. 4) The hydraulic head has been determined using the flow numerical model
(vector **h**) and the values of the objective function has been determined using the equation ().
5) the values of $T_{eq}$ are updated for each pilot points.
Using Levenberg–Marquardt algorithm the values of $T_{eq}$ for each pilot points is updated as long
as the objective function is minimized.
Figure 9 shows the results obtained from the flow model in steady state condition, calibrated
with the measurement campaign of February 2012 (Table 1).
Table 2 shows the data of model calibration and Figure 10 shows the graph of the calibration.
The outcomes of the calibration are satisfactory. The comparison between the simulated and
observed datum has given a mean absolute residual equal to 0.57 m, an RMSE equal to 4.57 m,
a correlation coefficient r$^2$ equal to 0.997. In the following figures and tables are shown the
results for the flow model.





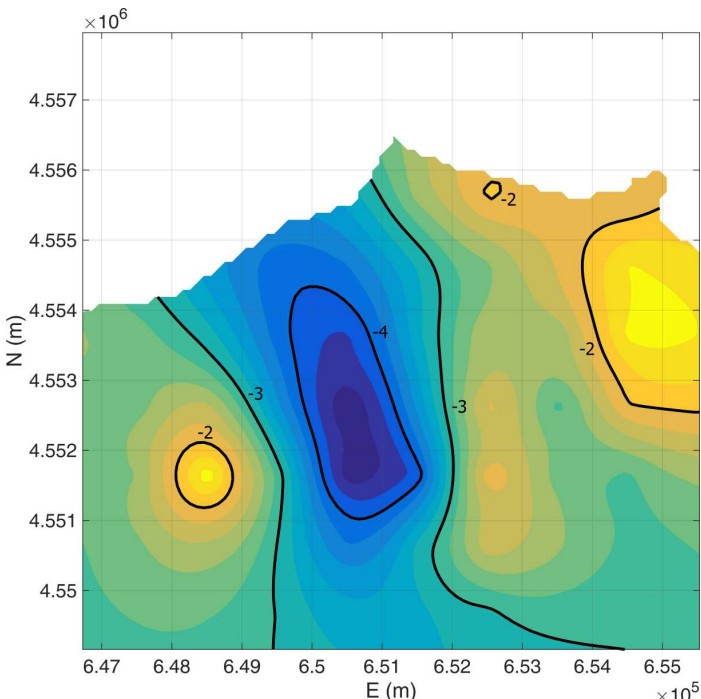


**Figure 8. Map of log10 of aquifer transmissivity determined by means of inverse modelling algorithm.**

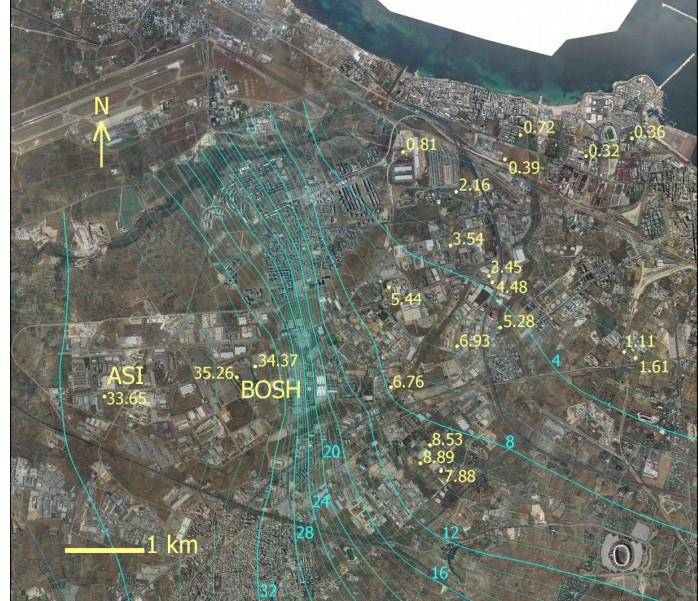


**Figure 9. Map of simulated hydraulic heads (blue line)**



| Name | Obs. Head (m) | Computed Head (m) | Residual Head (m) |
|------|---------------|-------------------|-------------------|
| P10 | 4.480 | 3.682 | -0.798 |
| L1-S | 5.278 | 4.835 | -0.443 |
| P11 | 1.611 | 2.205 | 0.594 |
| P19 | 1.110 | 2.217 | 1.107 |
| P14 | 0.321 | 0.515 | 0.194 |
| L2-S | 0.722 | 0.466 | -0.256 |
| P4 | 0.386 | 0.801 | 0.415 |
| L3-S | 2.163 | 1.870 | -0.293 |
| P3 | 5.441 | 5.519 | 0.078 |
| P16 | 3.536 | 3.315 | -0.221 |
| L4-S | 3.450 | 3.567 | 0.117 |
| P18 | 6.926 | 5.851 | -1.075 |
| L5-S | 33.649 | 35.587 | 1.938 |
| L8-S | 8.532 | 8.809 | 0.277 |
| L7-S | 7.880 | 9.516 | 1.636 |
| L6-S | 8.892 | 9.651 | 0.759 |
| P13 | 0.807 | 1.281 | 0.474 |
| L9 | 0.705 | 0.236 | -0.469 |
| L10 | 0.167 | 0.276 | 0.109 |
| L11 | 0.317 | 0.279 | -0.038 |
| L12 | 0.360 | 0.245 | -0.115 |
| L13 | 0.418 | 0.144 | -0.274 |
| L14-S | 34.370 | 33.776 | -0.594 |
| L15-S | 35.260 | 34.477 | -0.783 |
| P2 | 6.760 | 7.865 | 1.105 |

**Table 1. Comparison between the observed and simulated hydraulic heads with related residual, relatively to the measurement campaign of February 2012.**


| | |
|---|---|
| **Mean Residual** | -0.138 |
| **Mean Absolute Residual** | 0.566 |
| **Root Mean Squared Residual** | 0.743 |
| **Sum of Squared Weighted Residual** | 4.571 |


**Table 2. Data of model calibration.**



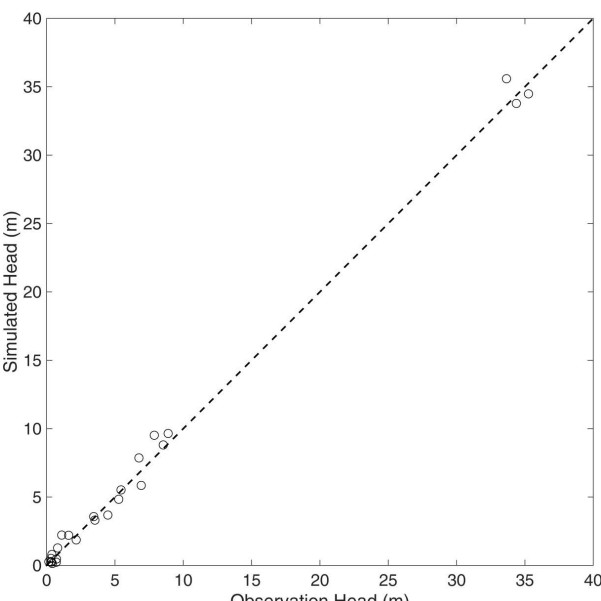


**Figure 10. Graph of the calibration.**


Once obtained the equivalent hydraulic transmissivity map and assuming a values of the
number of set of fractures $n_f$ the spatial distribution of the mean equivalent aperture and the
velocity field can be obtained.
Assuming valid the cubic law the mean equivalent aperture can be obtained as:
$$b_{eq} = \sqrt[3]{12 \frac{T_{eq}}{n_f} \frac{\mu}{\rho g}}$$                    (22)
The velocity field resulting:
$$v_x = -\frac{\rho g}{\mu} \frac{b_{eq}^2}{12} h_x$$
$$v_y = -\frac{\rho g}{\mu} \frac{b_{eq}^2}{12} h_y$$
             (23)

Whereas assuming valid the Darcy – Weisbach equation the mean equivalent aperture and the
flow field can be obtained by means of the following iterative steps starting from the values of
$b_{eq}$, $v_x$ and $v_y$ previously evaluated:





$$\mathrm{Re}^k = \frac{\rho \left| v^k \right| 2b_{eq}^k}{\mu}$$

$$f^{k+1} = \frac{123}{\mathrm{Re}^k}\left(1 + 0.12\left(\mathrm{Re}^{k\,0.687}\right)\right)$$

$$b_{eq}^{k+1} = \sqrt[3]{\frac{T_{eq}}{n_f^2}\frac{f^{k+1}}{4g}\left|\nabla h\right|}$$

530 (24)

$$v_x^{k+1} = -\frac{\sqrt{\dfrac{4b_{eq}^{k+1}}{f^{k+1}}g}}{\sqrt{\left|\nabla h\right|}}h_x$$

$$v_y^{k+1} = -\frac{\sqrt{\dfrac{4b_{eq}^{k+1}}{f^{k+1}}g}}{\sqrt{\left|\nabla h\right|}}h_y$$

Figure 11 shows the relative percentage of error on the flow velocity for different number of
fractures.

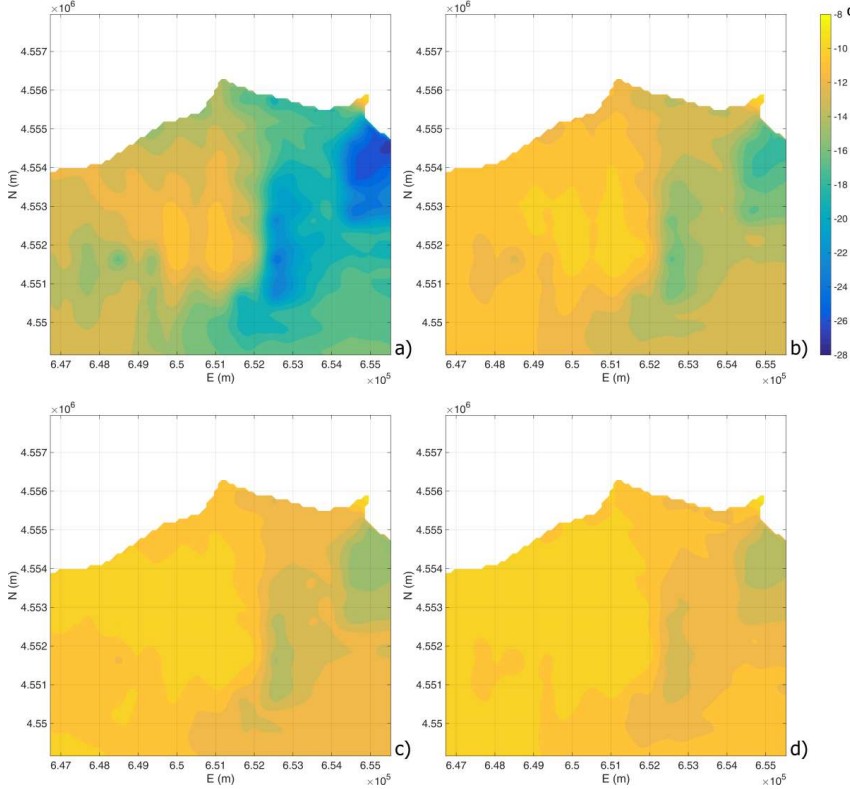




**Figure 11. Relative percentage of error on the flow velocity for different number of fractures: a) $n_f$ = 4; b) $n_f$ = 12; c) $n_f$**
**= 20; d) $n_f$ = 28.**

**Detection of the sources of contamination**

A particle tracking transport method has been applied for the simulation of contaminant
transport. The obtained simulation scenario proves to be compatible with the observed one and
therefore it is possible to assume that the sources of contamination are located in
correspondence of the observed hot spot (Figure 12).

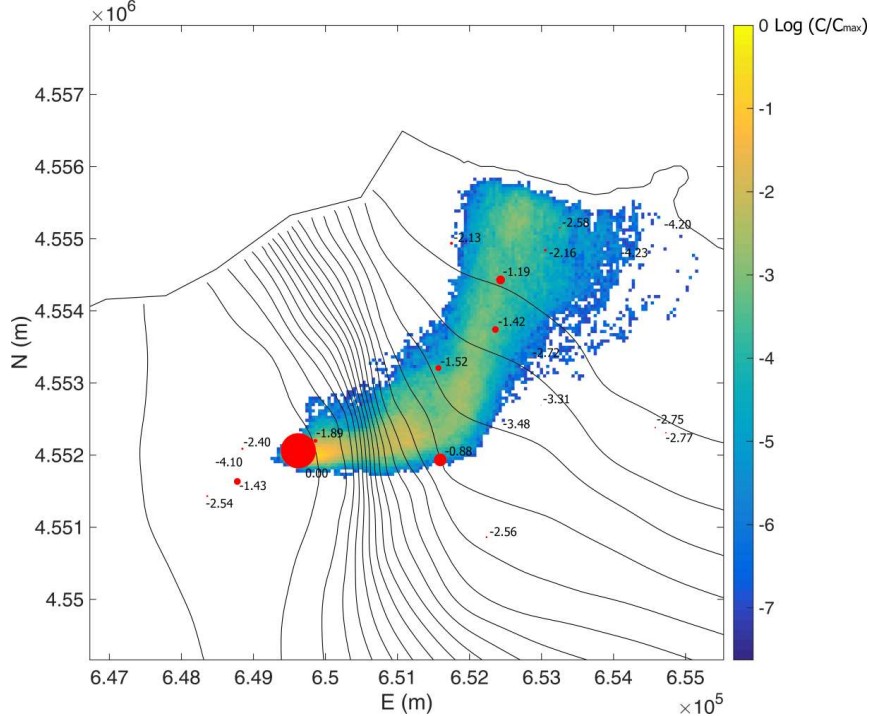

**Figure 12. Steady state distribution of hypothetical contamination using the random walk model with the source**
**contamination localized in correspondence of the hot spot of the contamination considering a number of fracture of $n_f$**
**= 20 and a longitudinal and transversal dispersion coefficient equal to $\alpha_L$ = 70 m and $\alpha_T$ = 7 m.**
Figure 13 shows the breakthrough curves of hypothetical continuous contamination released in
correspondence of the hot spot, determined for linear and nonlinear flow model, evaluated at
the downstream boundary for $n_f$ = 20. The nonlinear model shows a delay in the breakthrough
compared with the linear one. This is coherent with what detected by Cherubini et al (2014)
who found a delay in advective solute transport for a nonlinear flow model in a fractured rock
formation respect to the linear flow assumption.





In fracture networks, the presence of nonlinear flow plays an important role in the distribution
of the solutes according to the different pathways. In fact, the energy spent to cross the path
should be proportional to the resistance to flow associated to the single pathway, which in
nonlinear flow regime is not constant but depends on the flow rate. This means that by changing
the boundary conditions, the resistance to flow varies and as a consequence the distribution of
solute in the main and secondary pathways also changes, giving rise to a different behavior of
solute transport (Cherubini et al. 2014).
Figure 14 shows the mean travel time at varying number of fractures for the linear and nonlinear
model. With increasing number of fractures, the travel time increases in a linear way, because
the cross section area increases as well. The travel time for the nonlinear model is higher than
the linear assumption, coherently with the previous finding.

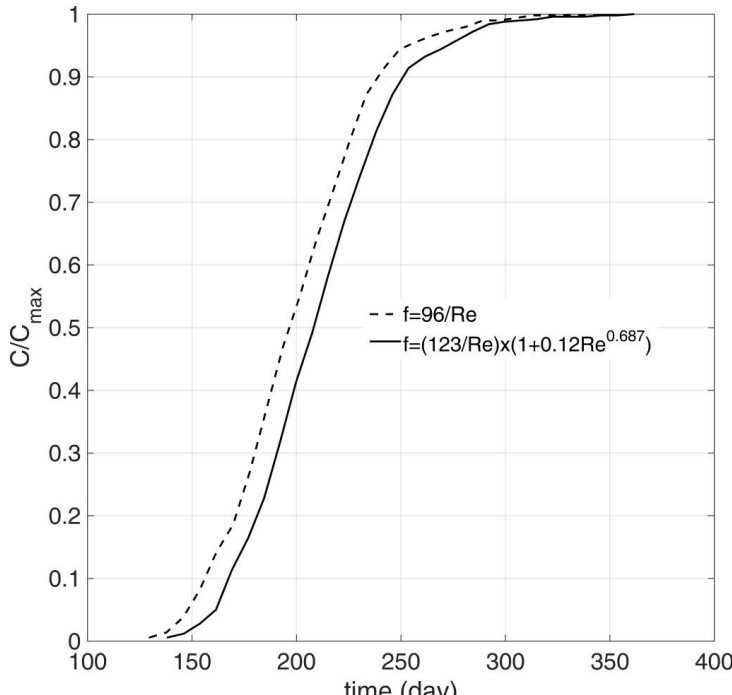


**Figure 13. Breakthrough curves of hypothetical continuous contamination released in correspondence of the hot spot,**
**determined for linear and nonlinear flow model, evaluated at the downstream boundary.**



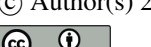

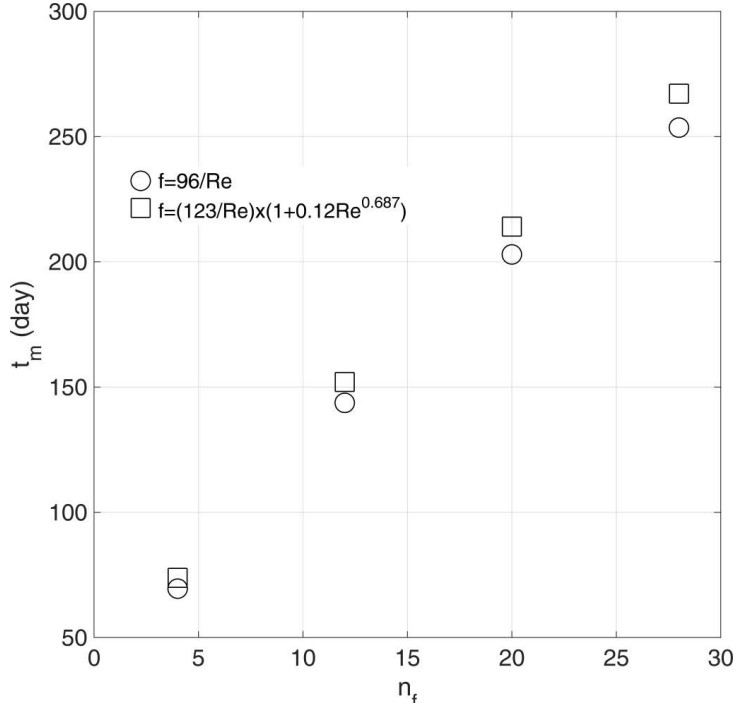


**Figure 14. Mean travel time $t_m$ at varying the number of fractures $n_f$ for linear and nonlinear model.**

## Conclusions

The present study is aimed at analysing the scenario of groundwater contamination (by
investigating the hotspots) of the industrial area of Modugno (Bari –Southern Italy) where the
limestone aquifer has a fractured and karstic nature.
The presence of hot spot areas has been detected, located upstream of the groundwater flow,
coherently with the state of contamination detected downstream.
A rough walled parallel plate model has been implemented and calibrated on the basis of
piezometric data and has coupled a geostatistical analysis to infer the values of the equivalent
aperture. Using the random walk theory, the steady state distribution of hypothetical
contamination with the source contamination at the hot spot has been carried out.
The flow and transport model have well reproduced the flow pattern and have given a pollution
scenario that is compatible with the observed one.
From an analysis of the flow and transport pattern it is possible to infer that the anticline
affecting the Calcare di Bari formation in directions ENE-WSW influences the direction of flow
as well as the propagation of the contaminant.



The results also show that the presence of nonlinear flow influences advection, in that it leads
to a delay in solute transport respect to the linear flow assumption. Moreover, the distribution
of solute according to different pathways is not constant but is related to the flow rate
This is due to the non-proportionality between the energy spent to cross the path and the
resistance to flow for fractured media, which affects the distribution of the solutes according to
the different pathways.
The obtained results represent the fundamental basis for a detailed study of the contaminant
propagation in correspondence of the hot spot area in order to find the best clean up strategies
and optimize any anthropic intervention on the industrial site
Future developments of the current study will be to implement a transient model and to include
the density dependent flow into the simulations.

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
