# Peer review of "Numerical modelling of flow and transport in Bari industrial area by means of"

_Hydrology and Earth System Sciences, 2018_

## Referee Comment (RC1) · N. Sund (Referee) · 19 May 2018

The manuscript is well structured and presents very strong ties to prior work. The modeling methods are generally well explained and well suited to the case study presented. There are a few places within the manuscript where further clarity would help the authors' message.

1. Line 98-99 "...each time applying v based on one of the three 'equivalent apertures'.": I think the authors are referring to v as velocity, however, introducing variables in the introduction could lead to confusion.

[Figure]

2. Line 399: $n_f$ is not yet defined here.

3. Line 457 "Where \alpha_L(L) and \alpha_T(L) are the longitudinal and transverse dispersion coefficients respectively.": These are the dispersivities, not the dispersion coefficients.

4. Line 459: "The solute plume is discretized into a finite number of particles.": How many?

5. Line 465: $Z_1$ and $Z_2$ are not defined. Also, $Z_1$ and $Z_2$ for the x and y jumps should be different from each other.

6. Line 468-470 "For steady-state flow and for a source constant intensity, the assumption that the particles N released in time interval (t1,t1+\Delta t) follow exactly the same random trajectories of the particles N released during the previous interval (t_1,t1-\Delta t) is possible.": Why is this true? Please add a reference to back this claim up. Since your random walk method should be drawing new random numbers every step for every particle, it seems that this should not be true.

---

## Referee Comment (RC2) · E. Zechner (Referee) · 4 Jun 2018

The manuscript under discussion presented by Claudia Cherubini et al. combines existing concepts for modeling flow and transport in fissured/karstified aquifers, which in the past have been mostly used for theoretical studies or laboratory experiments, to characterize a contaminated site in a limestone aquifer of Southern Italy. This applied part in their work clearly presents a benefit to the scientific community. However, the manuscript would benefit from clarifications from the authors on the novelty in their work, on stating how certain observations/data are essential for the study, on a balanced discussion of the comparison simulated vs. observed data, on improved figures

illustrating the simulation results, and on a revision of the language and grammar.

Their work is introduced by an extensive overview, or almost review of the existing concepts to simulate flow and transport in fissured/karstified aquifers. In my opinion, this section could be slightly more focused/shortened to existing research which is in direct relation to the presented work. The section would benefit from a statement where the presented work is situated compared to the existing work, and where it presents a novel approach. The geological framework is described in some detail, although is not clear why some of the details such as depositional environment of the lower Pleistocene on top of the Bari Formation, presence of syn-/anticline axis, the geophysical properties (P-, S-Wave velocity), or geoelectrical surveys are useful for the study. In addition, properties of the Bari Fm are presented in the Lower Pleistocene part such as the hydraulic conductivities from 10E-3 to 10E-4 m/s. It is also not stated that this range likely presents an average, assuming that the value range would be much wider in a karstic aquifer? Data of hydrology and hydraulics, data interpolation and model implementation is straight-forward, and step-by-step. However, improved visibility of illustrations and figure legends would create benefits, e.g. show mentioned two hotspots in Fig. 6, including naming substance displayed (PCE?); blue labels in Fig.9 are not visible; Fig. 11 is to small, and its results are not mentioned or discussed in the text(?): what are the flow velocity errors exactly, errors compared to which velocity, darcy-flow velocities without fractures? In Fig. 12 simulation results are compared to normalized measurements, but figure resolution/size make it difficult to compare The manuscript would generally benefit from more extensive discussion of the results, e.g. in a separate discussion chapter.

L(ine) 25: . . .WITH respect. . .to the NON-constant. . .

L72ff: avoid one sentence paragraphs

L199: ..consider pulse-like. . .

L228: The formation shows. . . (no paragraph)

L232: unclear what "not interested by tectonical discontinuities" means

L260: how can the presence of a fault line control the development of the actual hydrographic network?

L267: Average of K-values? (see comment above)

L268: What does "under low pressure" mean, under a low GRADIENT?

L271: what is the (hardly visible) hydrographic network in Fig. 1? An area of climatic stations?

L300: Rather "estimated", or "calculated" EVT instead of "real"...

L495: equation number missing

L540: L374 mentions two hotspots, here only one observed?

Please revise grammar/language

———————————————

---

## Author Comment (AC1) · 2 Jul 2018

Rev 1 1. That paragraph has been removed, because the introduction has been judged as too long, according to reviewer n° 2: "In my opinion, this section could be slightly more focused/shortened to existing research which is in direct relation to the presented work". 2. nf has been defined. 3. 'dispersion coefficients' has been changed with dispersivities. 4. The number of particle are 500 for each time step. 5. $Z_1$ and $Z_2$ has been defined and they are different for x and y jumps. 6. The reference has been added Rauch et al. 2005.

106, 2018.
* * *
Interactive
comment

---

## Author Comment (AC2) · 2 Jul 2018

The introduction has been significantly reduced, and a part has been added where the novel approach introduced by the manuscript has been added at the end of the introduction. The importance of the lower Pleistocene, syn-/anticline axis and geophysical properties has been added in the text: "The degree of fracturing degree affecting of the Calcare di Bari formation is quite variable and mainly depends mainly on the geological and structural (faults, anticline axis,...) evolution of the area including faulting and folding. Also the distribution of the local measurement of the Rock Quality Designation (RQD) index is confirmed by the variability of the electrical resistivity along geoelectrical profiles (with length from 500 to 1000 m) and from the propagations of the P and S waves (seismic measurements; length of about 1000 m)." As far as the variability of the hydraulic conductivities, several studies indicate that in some carbonate aquifers where karstic phenomena are present, the hydraulic conductivity (k) is scale dependent (Galvão et al., 2016; Király, 1975; Sauter, 1991; Halihan et al., 2000). In particular, the hydraulic conductivity at the small scale ($10^{-2}$-1 m) shows values of $10^{-8}$-$10^{-5}$ m/s, due to the presence of microfissures and the matrix effect, while at a larger scale (1-100 m) typical values range from $10^{-7}$ to $10^{-3}$ m/s as a consequence of the macrofissures effect. At an even larger scale (i.e. regional, 100-$10^4$ m) k slightly increases to $10^{-5}$-$10^{-2}$ m/s because of both fracture and karstic effects. These variations are mainly a consequence of the regional distribution and size of brittle deformation structures, like faults and joints, and localized karstic phenomena affecting the aquifer. Galvão P., Halihan T., Hirata R. (2016): The karst permeability scale effect of Sete Lagoas, MG, Brazil. Journal of Hydrology 532, 149–162. Király L. (1975): Rapport sur l'état actuel des connaissances dans le domaine des caractères physique des roches karstique. In: Burger, A., Dubertet, L. (Eds.), Hydrogeology of Karstic Terrains. International Association of Hydrogeologists, Paris, pp. 53–67, Series B, No. 3. Sauter M. (1991): Assessment of hydraulic conductivity in a karst aquifer at local and at regional scale. In: Proc. Third Conference on Hydrogeology, Ecology, Monitoring and Management of Ground Water in Karst Terranes, December 1991, Nashville. Halihan T., Sharp J.M.Jr., Mace R.E. (2000): Flow in the San Antonio segment of the Edwards Aquifer: matrix, fractures, or conduits? In: Wicks, C.M., Sasowsky, I.D. (Eds.), Groundwater Flow and Contaminant Transport in Carbonate Aquifers. Balkema, Rotterdam, The Netherlands, 129–146. Fig. 6 the caption has been modified by adding PCE. Fig. 9 has been modified in order to improve the visibility of the piezometric head contour line. Fig. 11 has been changed In order to improve the visibility by showing only the percentage of error for nf equal to 4 and 28 respectively. Fig. 12 the visibility of simulated plume has been improved and so the qualitative comparison between the observed and simulated values becomes more straightforward. The paragraph "Discussion and

results" has been added including the following sub paragraph: "flow modeling", "detection of the sources of contamination". Some more paragraphs have been added to discuss the results. L(ine) 25 the modification has been done. L72ff One sentence paragraphs have been put together. L199 'Also' has been removed as requested (so the sentence is 'consider pulse-like'). L228 the typesetting error has been corrected. L232 'not interested by tectonical discontinuities' has been substituted with 'does not show tectonical discontinuities'. L260 the answer to this point has been added in the text: "As regards the structural features of these deposits it is possible to observe that the anticline affecting the Cretaceous succession of the Calcare di Bari formation with an ENE-WSW axial direction (Fig. 1) causes a partial diversion of the water courses, whose path seems to be also influenced by some NE-SW fault (NE of Modugno). The former phenomenon is due to the antithetically dipping flanks of the gentle fold, while the latter effect is likely a consequence of the denser fracturing along the shear zone and hence the increased erodibility of the local outcropping limestone enhancing the water flow concentration. In general, the limestone bedrock hosts a wide and thick aquifer due to a diffuse rock fracturing and the karstic phenomena." L267 Average K values are from 10-3 to 10-4 as written in the text: "The average hydraulic conductivity of this aquifer is generally estimated in $10-3$ to $10-4$ m/s." L268 Under low pressure has been changed with low gradient L271 'hydrographic network' has been substituted with "flow pattern of ephemeral and intermittent streams. " L300 Real has been changed with estimated L495 The number of equation has been added L540 the comment is right. The correction has been done. 'The presence of one hot spot has been detected, located upstream of the groundwater flow, coherently with the state of contamination detected downstream.'

---

## Author Response (AR1)

**1. The observed high hydraulic heads has to be better explained (L359-L366). It cannot be due to the zone of poor connection only.**

The position of the escarpments and of the anticlinal axis has been inserted in the piezometric map (see figure below).

[Figure]

What can be seen is:

   a) The anticlinal behaves as a local subterranean watershed;
   b) The lower hydraulic head values (5-8 m, asl) are downstream of the escarpments.

[Figure]

Profilo lungo la linea rossa.

Downstream of the escarpments there is the Calcarenite di Gravina formation, whereas upstream (BOSCH and ASI areas) the Calcare di Bari formation; a possible explanation for the increase in hydraulic gradient is: 1) lower transmissivity as showed through the step drawdown tests; 2) the transition from a more permeable

outcropping lithotype to a less permeable one resulting in a decrease of the effective infiltration; 3) hydraulic disconnection due to lower interconnectivity of the fracture system.

**2. The flow model is calibrated in steady state on data collected in Feb. 2012. Are these data representative of steady state?**

The data collected in Feb 2012 reasonably show a regional trend for water table height, orientation, flow path distribution and gradient. Of course a monthly collection of hydraulic head data may improve the estimates of aquifers properties and the simulation of hydraulic head and contaminant transport.

**3. There is salt water intrusion at the coast boundary and it seems you neglected it in the modelling. Why?**

In the study area groundwater is floating on an underlying saltwater zone. It has been involved by the process of seawater intrusion showing a concentration of salt variable from 0.5 and 2.5 g/l (Cherubini, 2008). Anyway this salt concentration gives rise to a small effect on the density of freshwater. Therefore it is reasonable to assume a constant density model neglecting the saltwater intrusion.

**4 The transport seems to be simulated in 2D. Are you sure that the concentration at the source is homogeneous over the all depth of the aquifer to allow this approximation? Are the measured concentrations representative of the average concentration over the depth?**

The groundwater flows through a sub parallel fractured layer separated by compact rock blocks. In the study the stratigraphic sequence observed during drilling of wells, from bottom upward, is Jurassic dolomites (with a thickness of 21m), Cretaceous limestone (with a thickness of 26m), and Pleistocene sandstone (with a thickness of 5m) (Masciopinto et al. 2013). The whole thickness of the fractured layer where groundwater flows varies in the range of 10 – 30 m, whereas the field scale of the horizontal flow is of the order of magnitude of $10^4$ m. The source of contamination is located in an industrial area where in the past chlorinated solvents have been spilled directly into the groundwater wells. For this reason a two dimensional flow and transport and point source contamination assumption can be considered reasonable. The concentration has been monitored into the groundwater wells using a bailer. In order to obtain the representative aquifer conditions, purging immediately prior to sampling has been carried out.

**5. The subtitle L356 is misleading. How did you detect the sources?**

The paragraph at L356 doesn't talk about sources but it is about the' Analysis of the scenario of contamination for the study area'.

The subtitle 'Detection of the sources of contamination at L533 has been changed into '*Analysis of the scenario of contamination*' which is no more misleading.

**6. The most innovative part of the work is the comparison between linear and non-linear flow. (Fig. 13). Please provide more information and highlight the differences between Cherubini et al (2014) work.**

The following periods:

"Figure 13 shows the breakthrough curves of hypothetical continuous contamination released in correspondence of the hot spot, determined for linear and nonlinear flow model, evaluated at the downstream boundary for nf = 20. The nonlinear model shows a delay in the breakthrough compared with the linear one. This is coherent with what detected by Cherubini et al (2014) who found a delay in advective solute transport for a nonlinear flow model in a fractured rock formation respect to the linear flow assumption."

Have been substituted with:

"Figure 13 shows the breakthrough curves of hypothetical continuous contamination released in correspondence of the hot spot, determined for linear and nonlinear flow model, evaluated at the downstream boundary for nf = 20.

Figure 14 shows the mean travel time at varying number of fractures for the linear and nonlinear model. With increasing number of fractures, the travel time increases in a linear way, because the cross section area increases as well. The figures highlight that travel time for the nonlinear model is higher than the linear assumption. In particular way the percentages of error are in the range of 6.22 – 5.34 % passing from $n_f$=4 (Re = 0.02 – 10.60) to $n_f$=28 (Re = 0.002 – 1.51). This is coherent with what detected by Cherubini et al (2012, 2013, 2014) who carried out hydraulic and tracer tests on an artificially created fractured rock sample and found a pronounced mobile–immobile zone interaction leading to a non-equilibrium behavior of solute transport.

The existence of a non-Darcian flow regime showed to influence the velocity field by giving rise to a delay in solute migration with respect to the values that could be obtained under the assumption of a linear flow field. Furthermore, the presence of inertial effects showed to enhance non-equilibrium behavior. In particular manner they found that percentage of error on the travel time respect to the linear flow assumption varied in the range of 5.90 – 40.75 % corresponding to a range of Re of 29.48 – 52.16. These results highlight that as the scale of observation increases the error on the mean travel time respect to the linear flow model becomes more relevant. In fact, at field scale also for Re just above the unit (nf = 28) the error is equal to 5.34 % comparable with the error of 5.90 % found at laboratory scale for Re equal to 29.48. This means that under anthropic stresses multiple pumping or injections give rise to a higher flow velocity and then higher Re leading to a dramatic delay on contaminant transport. Therefore, nonlinear flow must be considered in order to have a more accurate estimation of the breakthrough curve and mean travel time of contaminated scenarios. "

---

## Author Response (AR3)

Answer to Editor

Q1: The local high value of the piezometric head cannot be due to heterogeneity only. There should be some additional recharge.

The presence of sinkholes and fissures at surface can give rise a point source recharge and then can contribute to the presence of the local high value of piezometric head.

The following sentence has been added to the manuscript:

'3) presence of sinkholes and fissures at surface giving rise a point source recharge'

Q2: I am asking you if the data are representative of steady state conditions. Of course, they will show a trend but they may differ from steady state conditions.

The data are representative of steady state conditions because it is reasonably assume that the magnitude and direction of flow does not change with the time.